# Lived Experiences of Domestic Violence in Women and Their Children: A Phenomenological Study

**DOI:** 10.3390/healthcare10081556

**Published:** 2022-08-17

**Authors:** Pei-Yu Lee, Bih-O Lee

**Affiliations:** 1Department of Early Childhood Care and Education, Cheng Shiu University, No.840, Chengcing Rd., Niaosong Dist., Kaohsiung City 833, Taiwan; 2College of Nursing, Kaohsiung Medical University, Kaohsiung City 807, Taiwan

**Keywords:** children, domestic violence, qualitative research, lived experience, woman

## Abstract

This study explores women and their children’s lived experience of domestic violence (DV). A qualitative phenomenological research approach was used. Data were collected by semi-structured interviews. Five women and five adult children participated in this study. COREQ reporting guidelines were utilized. Three main themes and six subthemes emerged from the interviews with the women; the main themes were “living with suffering”, “insecurity in daily life”, and “conformity in coping behaviors”. Two main themes and four subthemes were identified from the interviews with the children; the main themes were “barriers to learning and interactions with peers” and “a sense of threat to life”. The findings show that the women and their children had to cope with not only the DV itself but also the negative consequences of the violence. Several recommendations are made for the police and first-line healthcare and social work systems and to prevent DV by integrating the care provided to the family. The findings provide prevention and integration care for mothers and their children experiencing DV systematically.

## 1. Introduction

Domestic violence (DV) is violence or abuse by one intimate partner against another in a domestic setting. DV includes physical and sexual violence and social, economic, verbal, emotional, and spiritual abuse [1]. Taiwan’s Domestic Violence Prevention Act (DVPA), passed in 1998 and amended in 2015, was the first set of such laws and regulations in Asia. The DVPA is applicable when violence occurs between two people in a formal marriage or a relationship equal to a husband-and-wife partnership.

DV occurs globally. According to the National Coalition Against Domestic Violence, the DV statistics for the US showed that from 2016 to 2018, the number of intimate partner violence victimizations increased by 42% [2]. About 20 people per minute are physically abused by an intimate partner [3]. In Australia, by the age of 15, around 1 in 14 (6.9%) children experience physical abuse by a family member. In addition, in 2016, 1 in 20 (5% or 117,000) women aged 18–34 experienced intimate partner violence, compared to 1 in 66 (1.5% or 96,000) aged 35 and over [4]. In Taiwan, the Ministry of Health and Welfare reported that there were 137,148 suspected cases of DV in 2017 and the number of these cases increased by 19,986 between 2011 and 2017 [5]. However, statistics are lacking for children witnessing DV in Taiwan; the number of children receiving child protection services is around 5000 per year. In addition, a recent study showed that there is an increasing number of women facing DV during COVID-19 lockdowns, indicating DV-related issues are of great concern and should be studied domestically and internationally [6].

## 2. Background

The core characteristics of DV is chronic, long-term, and repeated abuse. Women who experience long-term DV throughout life will show various physical and psychological symptoms. For instance, the physiological symptoms could include sleeping disorders, musculoskeletal pain, headaches, insomnia, fatigue, stress, and other phenomena. Psychological symptoms include fear, anger, resentment, depression and phobia, a feeling of inferiority, low self-esteem, and severe self-injurious behavior [7]. Children who witness DV incidents are at greater risk of serious health problems in adulthood, including obesity, cancer, heart disease, depression, substance abuse, and tobacco use, as well as unintended pregnancies, than peers who do not witness DV [7,8]. The findings of Hsieh, Feng, and Shu, who analyzed in-depth interviews of eight abused women from southern Taiwan, showed that the women regretted their marriages, lived in unsafe environments, had economic difficulties, and endured unhappy lives [9]. In Shen’s study of young adults, most participants had experienced parental marital violence and child maltreatment since early age and the results also showed that child abuse was a significant corollary of DV. The likelihood of child abuse is higher in households where DV has previously occurred [10]. Thus, in Taiwan, the Child and Youth Welfare Law No. 43 of 2010 stated that children and adolescents who have witnessed DV must join an intervention program and be provided further counseling.

Only one journal article based on quantitative research on data for Taiwan has been published [11]. For other countries, a search of the databases, CINAHL, EBSCO, and PubMed, for the period 2010–2020, found 79 articles in total; over 85% presented quantitative research, which was mainly used to develop measurement tools, treatment systems, reporting procedures, etc. [12,13], but qualitative interviews were relatively rare [14]. More recently, one study was conducted in Puerto Rico, interviewing women who experienced intimate partner violence to get their perceptions of DV. The findings showed that there was a lack of disclosure and norms in Hispanic cultural values. Women with DV did not have knowledge of resources available to help themselves [15]. To deeply understand the phenomena associated with women and their children’s lived experience of DV, the aim of this study was to explore the DV experience of women and children to get insights for their healthcare or social care.

## 3. Methods

### 3.1. Study Design

This study used descriptive phenomenology to understand the DV experience of women and their children. Descriptive phenomenology, a qualitative inductive research approach, is particularly suited to complex experience or feelings that are difficult to quantify [16]. The COREQ guidelines were utilized.

### 3.2. Setting and Participants

Purposive sampling was used to select participants from a private community group called Women’s New Knowledge Foundation (WNKF). The WNKF is a nonpartisan, not-for-profit legal and legislative advocacy organization fighting for gender equality and women’s empowerment in Taiwan. The prerequisites for participation included: (a) in the case of abused women, that they were victims of DV in the past and can communicate in Chinese or Taiwanese; (b) in the case of the abused women’s children, that they had witnessed DV during childhood or adolescence, have reached adulthood (20 years of age), and can communicate in Chinese or Taiwanese and clearly and unambiguously express themselves in words; and (c) that both women and children agreed to be interviewed. Exclusion criteria included having a mental illness, emotional disorder, etc. (as diagnosed by a physician).

### 3.3. Data Collection

A preliminary study was used to test the feasibility of the main study before formal data gathering. An expert was invited to discuss difficulties in revising the interview guidelines. The researcher who was in charge of interviews has a lot of experience in conducting qualitative studies. The participants were free to choose the interview venue to ensure that the interviews were conducted in a safe environment such as home or workplace. Data were collected from December 2016 to June 2017. Data collection ceased when no new themes emerged. The interviews were conducted 1 to 2 times for 60 to 90 min face to face. The five women participants were coded A, B, C, D, and E and the five children were coded F, G, H, I, and J.

This study adopted in-depth interviews for data collection, with semi-structured interviews forming the basis of interview reference [17]. The interview questions for the women participants were: (a) Talk about your family, the ways in which you communicate with your spouse, and how power is shared in the household. (b) How do parents and children communicate and interact with one another? (c) Talk about your experience of DV including the present situation, the process of asking for help, and the children’s reaction. (d) After DV occurred, how did you face and adjust to it? (e) After DV occurred, how did the interactions change between family members? The interview questions for the participating children were: (a) Talk about the ways in which your parents interact with each other. (b) How do your parents communicate in the house? (c) Talk about the experience of witnessing the abuse your mother suffered including the present situation, the process of asking for help, the mother’s reaction, etc. (d) After domestic violence occurred, how did you face and adjust to it? (e) After DV occurred, how did the interactions change between family members?

### 3.4. Data Analysis

After obtaining research agreement from the interviewee, face-to-face in-depth semi-structured interviews began. The researcher maintained a neutral attitude and showed empathy when responding to the participants without any preconceived notions and criticism to avoid the involvement of subjective value judgments. The data were analyzed immediately, by debriefing after each interview and by listening to the audio recording and verifying the data. All the taped interviews were read and listened to several times to ensure that participants’ significant experiences were noted. Clusters of themes were identified and descriptions of coping with DV were recorded as exhaustively as possible. Moreover, a final validating step was performed by returning to each participant to go through the whole interview findings again. The interview content was divided into two parts: women and children.

### 3.5. Rigor

The rigor of this study was underpinned by Lincoln and Guba’s criteria [18]. (a) Credibility: two informal telephone interviews were arranged to establish a trusting relationship before formal interviews; furthermore, a reflective journal was used to increase the richness and credibility of the data. (b) Transferability: interviews were accurately and truthfully transcribed verbatim for presentation in this study. Transcription of interview content returned to participants for correction. (c) Dependability: we invited two nursing professionals with wide experience of qualitative research to review and modify the classification of the findings. (d) Confirmability: the researchers safeguarded all the reflective field notes and records of data analysis in this study for future verification and reference.

### 3.6. Ethical Considerations

Approval for this study was obtained from a qualified hospital ethics committee (No.104-166) of the Institutional Review Board (IRB). Prior to participating in the study, the purposes of the study and the interviewing and recording process were explained to the participants and their written consent was obtained. It was made clear to the participants that they had the right to refuse to answer a particular question if they were unwilling to and the right to terminate or withdraw from participation in this study without any consequences. All information from the interviewees was confidential and was coded and stored in a secure cabinet. All participants remained anonymous when the study outcomes were published.

## 4. Results

Ten participants were interviewed (five women and five children). Eighty percent of the participants were female and only two of the children were male. The women’s average age was 59.2 years; they had suffered DV for nearly six years and had received counseling therapy for at least two years. Most of the women had been divorced for more than seven years, while only one woman remarried, five years after her divorce. The children had an average age of 25.6 years; they had witnessed DV for at least 4.5 years and had received counseling therapy for at least six months. Several themes and subthemes emerged from the interviews (Table 1).

### 4.1. Women

#### 4.1.1. Theme 1: Living with Suffering

The victimized women’s subjective feelings of physical pain and emotional weakness due to their long-term exposure to DV caused suffering in their lives. This theme included two subthemes, as follows.

##### Physical Pain

Women suffering from DV experienced a life of physical injury and pain. In the household, the unstable emotional explosions of spouses, which usually involved physical and verbal attacks, led to pain from bodily injuries. The women’s lives were full of stress. For example, some women said that

There were several times when I argued with him, and he would start to take a swing at me. Most of the time I dodged, but I failed twice and ended up seeing a doctor because my ears were beaten and I had tinnitus. I wasn’t hospitalized, so I didn’t report him to the police. He used to do woodwork transporting, so he had great physical strength. When he became upset, he would slap my face or kick me and our child. Due to our incompatible personalities, we would start arguing after even a short conversation. Therefore, interaction between family members wasn’t so frequent (A2-104-137).

There was a period of time when he would start cursing directly after coming home. I never knew the reason why. The neighbors would come and complain because of the loud noise. When I mentioned this to him, he would scold me (C1-104-012).

##### Emotional Distress and Suffering from Ambivalence

This subtheme refers to the distressing emotions of women who were victims of DV as well as the contradictory feelings they experienced when facing a loved one who was also an abuser. In order to protect themselves from the negative emotional impact of the perpetrator, the participants spent much energy worrying about their own and their children’s safety. For example, some women said that

I feel like I have distressing emotions. Sometimes I can’t control my own feelings. I used to spend my days bawling my eyes out. When he was angry, he would yell swear words and verbally assault me by saying that I should get hit by a car and so forth. I flinched, and never dared to respond. Even though I was very angry, I would only have made myself even more upset. I didn’t know why I married him in the first place (C1-104-040).

When he came home, I saw him sitting alone in the living room. Sometimes I would want to have a chat with him, but my heart resisted it. I really had conflicting feelings when I faced him. These contradictory feelings were very strong at this period of time. He seems like a stranger to me (B1-104-053).

#### 4.1.2. Theme 2: Insecurity in Daily Life

The experiences and feelings of women who suffered from DV included psychological problems and emotional reactions caused by the long-term DV from their partners and these psychological reactions were the reasons for the women’s low mood. The second theme included two subthemes, as follows.

##### Dealing with Low Self-Esteem and Self-Comfort

Women experiencing DV believe that they are unwelcome, which, in turn, leads to low self-esteem. However, they still need to look after their children, which means that they have to find a way to constantly comfort themselves. Most of the women interviewed said that when they faced other family members, they also felt that they had an obligation to them, especially to their children.

Luckily, my children know that I am concerned about them. I just could not do many things for them. After all, when my husband was angry, he would also hit the children. I was scared that the neighbors would ask me about it. I felt very ashamed, and this led to not wanting to greet or interact with them. Because of my marriage I have certainly become more negative. Many endeavors I would start with negative thoughts. I feel like I am useless. I make my parents worry about me (D2-104-177).

I often comforted myself. My marriage might be rough, but whenever I encounter a problem, all the family members from my parents’ home would assist me; no matter if it is solvable or unsolvable. I am already very lucky, since some people don’t even have anyone to lean on (E2-104-117).

##### Heavy Sense of Loss

This subtheme refers to women not being able to control their own emotions and often spending their days in tears after being subjected to DV and undergoing unaccountable irritation, anxiety, and feelings of despair. The relations with the family were full of confusion and contradictions. Most women did not know how to get along with their husbands.

This marriage made me more pessimistic. I would look at things in a negative way. After all, I have lost all confidence in marriage. I went to a fortune teller who said this was my fate. I once lost two kilograms in one month. I didn’t have much of an appetite and I developed ulcer symptoms which I still suffer from now. If he was home, we would have a lot of psychological pressure. We were scared that he would scold us because of his bad temper. When he wasn’t home, my child and I would feel relieved (D1-104-068).

I was really in an awful mood, so I interacted with my child less and less. Usually, I wanted to be alone. My sister said she would look after my child for me (C2-104-126).

#### 4.1.3. Theme 3: Conformity in Coping Behaviors

The women who were abused knew that they could not live without their partners and families, because they needed to consider their children who were still young. Although family life was full of struggles, to maintain a complete family, the women tried to seek social resources and only in this way could they continue to live. The third theme includes the following two subthemes.

##### Seeking Social Resources

This subtheme refers to the women participants needing a reason to live bravely. Women sought out news of poor underprivileged groups in newspapers or magazines to comfort themselves so that they might stay strong. Women also sought out news to understand how to get help from the social system.

During the first year of the beatings, we went for counseling at the department of social welfare several time. However, he kept the same violent habits. I remember that we were busy during those days, so we didn’t go afterward (B2-104-199).

Afterward, there were several times when the beatings became worse, so I reported it to the police. I didn’t know that there was marriage counseling before violence happened, but I guess it wouldn’t have worked because of his stubborn personality. However, I appreciated a lot the police helping me (C2-104-148).

##### Living in the Moment

Most of the women did not know how to get along with their husbands. The feelings they had when they faced the abuser were pity and sympathy. Because of this, the women stayed with their husbands and they had to bear verbal abuse from the husbands.

I truly despaired at the time. I couldn’t see the future. Once, when I finished work and came home, he was throwing a temper tantrum. It drove me crazy, but I didn’t have a place to go, I told myself that my husband only yelling at me, no physical violence (E1-104-079).

I have often comforted myself. I am already very lucky compared to those refugees from Africa or those who were beaten to death by their own families. When I feel the urge to end my life, I would tell myself not to give up so easily (D2-104-111).

### 4.2. Children

#### 4.2.1. Theme 1: Barriers to Learning and Interactions with Peers

Given that the children were afraid that their family affairs would become known to their schoolmates and that others might see them differently, they automatically reduced their social interactions and scope. Children were also unable to concentrate on reading because of the lack of security at home, so they were often scolded at school due to unsatisfactory grades or deviations in behavior, leaving them not knowing what to do. The first theme includes two subthemes as follows.

##### Alienation from Peers

The inconsistent family interactions did not only diminish interactions with peers, but also caused emotional alienation. The children were afraid that their family affairs would be laughed at by peers and gossiped about by others. This caused them to lose trust among peers and led to emotional estrangement.

I didn’t chat with the classmates. There were several times when I didn’t participate in school activities. It doesn’t matter to me. I just didn’t feel like going. I often used to talk and chat with classmates. Now, I chat with them through a computer network like MSN or Facebook. The interaction isn’t so frequent (F1-104-032).

I was scared that the classmates would talk about family affairs, so I didn’t want to chat. I just didn’t have a feeling of fulfillment when I communicated with them, so my motivation to interact with classmates was low (J1-104-115).

##### A learning Gap Led to Frequent Punishment

The main reason for the teenagers’ learning gap was that nobody was concerned about their homework, so they did not care about their learning outcomes, such as grades. When the children’s parents were in a bad mood and saw grades that indicated poor performance, they often resorted to hitting or scolding. This made the child suffering DV unable to appreciate his/her own value. The children even deliberately stayed home and often took school leave in order to protect their mothers.

My scores were always bad, and I was never in the mood to study either. I was often unwilling to go to school. Anyway, I already didn’t like to study. Afterward, my mom was worried and wanted me to study at my aunt’s house at weekends (F2-104-169).

If I didn’t get good grades, I would be beaten as well as laughed at by others who said I was stupid. I was disgusted with this kind of life. There was a period of time when I preferred to stay in the school dormitory and not go home. At least I wouldn’t have to listen to their fighting (I1-104-097).

#### 4.2.2. Theme 2: A Sense of Threat to Life

Although children constantly witnessed DV incidents at home, most of them said that they had not suffered physical harm. However, they were spiritually afraid and wanted to avoid these verbal and emotional-violence-related stimuli. Therefore, they were listless and indifferent to life. The children expressed that they were very pessimistic and did not want to talk to others, preferring to lock themselves in their room to forget the complex world outside. The second theme includes two subthemes, as follows.

##### Monotonous Home Life

Due to inconsistent family interactions, the time the teenagers spent with family members changed. A prolonged period of family interaction did not show them how to solve their parents’ disputes. The teenagers developed some activities that they deemed appropriate, such as watching TV, playing online games, occasionally chatting with classmates online, wandering around, etc., to entertain themselves in their own rooms.

I have a brother who is two years older than me. He is quiet and doesn’t like to talk when he is home, and most of the time he stays in his own room. When my parents were arguing, we would go upstairs. My mom would also ask us to go to our room. We probably used the computer or watched TV the whole afternoon since nobody cared about us (I2-104-187).

If I had trouble, I would talk to my mom instead of my dad, because I was afraid that if he was in a bad mood, he would take it out on me. Thus, I would observe my dad’s mood first, and decided whether or not I should talk to him, but most of the time I would talk to my mom first (H1-104-048).

##### Pessimistic Thinking and Low Self-Esteem

Due to being battered and abused, adolescents would automatically adopt a subjective viewpoint and pessimistic thinking. They would then habitually tend to see the negative side of everything or instinctively think that they would be battered or abused by others. Because of long-term blame by family members who abused them when dissatisfied, they would always have pessimistic thoughts and engage in self-blame.

I was already humiliated because my classmates probably knew about my father. Because of my inferiority complex, I wasn’t happy at home. I always thought that I was inferior to others. Once I intended to run away from home with my mother to escape my father (F1-104-081).

I used to contemplate why my father was different from other fathers. I always adopted a morbid view of everything, and always ended up thinking the worst. How could I change these bad habits? I have never had the desire to commit suicide, but I secretly cried with my sister several times. I also saw my mother crying. I felt powerless as well as useless (G2-104-087).

## 5. Discussion

To the best of our knowledge, this is the first study that focuses on DV from the perspective of both women and their children. The findings of this study indicated that the women faced not only specific DV episodes but also physical and psychological harm after the violence. The children had to deal with not only the specific DV episodes but also the consequences of the abuse and family dynamics that accompany this type of violence. The findings from both perspectives should be important for health-care professionals.

Compared to previous studies, the findings of this study expand our understanding of DV experience of both mothers and adult children. We covered the experiences of women subjected to DV and children who witnessed DV under five themes. Our results showed that women’s interpretation of their DV experience was partly similar to that discussed in previous studies, but experience related to the specific local culture also emerged in this study [14,19]. Traditional Chinese views of upbringing, including the beliefs that “familial ugliness should not be publicized” and “persuasion to stay not persuade to leave” make it difficult for others to discover DV incidents and the witnessing of such incidents and to intervene early. Some experiences of the children witnessing DV in this study are partially similar to those from previous studies [9,20]; most children were dissatisfied with their home environment. Thus, community and teachers’ support, school treatment programs, and community shelters should be involved. If these support systems can be actively provided in a timely manner to children who have witnessed DV, and if the children can actively participate in the discussion of and decision-making process for solutions, their ability to cope with DV would be enhanced.

It is clear that DV has many physical and psychological consequences for women and children. In their everyday family life, in addition to work pressure, women must spend time dealing with issues of family harmony due to the violence of their husbands. The women in this study all had children; thus, they also had to pay attention to protecting their children. Previous studies have shown that women with children have more extreme fears and insecurities about their living environment compared to women without children [9]. In this study, women did try to call the Social Affairs Bureau or Department of Police for help, but it was in vain. Hence, they requested the government develop a project plan to enable mothers and children to take immediate refuge in a safe haven. Most women also visited hospital after a DV incident, but the health-care providers (doctors or nurses) were unable to immediately deal with their emotional problems [21]. This aspect should be emphasized by the health-care system and government.

Only one woman in this study actively sought help from her own family. The women only mentioned the DV to their friends or neighbors and sought other social support by themselves. For women experiencing DV, physical and mental recovery is a long-term effort. Physical violence is often accompanied by emotionally abusive and controlling behavior as part of a much larger, systematic pattern of dominance and control. The frequency and severity of DV vary dramatically [22]. This study also found that the possible reason for the extreme fear of the perpetrator expressed by women with children may be their unsafe living situation. The presence of children indicates that the victim and the perpetrator are more likely to be living together or staying in contact even after separation due to joint child custody. Consequently, these women may still be forced to keep in contact with their ex-husbands due to joint custody or custody disputes [23].

Most children witnessing DV react in a negative way in their emotions, leading to lack of self-confidence and fear of being in contact with other people. This study found that, due to long-term blame or battery, the children witnessing DV could not control these emotions and deal with the emotional entanglement between love and hate toward their family members, a result similar to that of Holt, Buckley, and Whelan [19]. Thus, long-term follow-ups may be important for underlying mental problems for children witnessing DV [24].

In this study, two children mentioned that they wished to call the police to actively provide solutions. Once the police receive the DV emergency report by phone, their attitude and management of the situation may influence the degree of physical and psychological damage to the child reporters. Some studies found that police decisions are important and that appropriate risk assessment and management by the police are necessary [25]. In addition, two children also mentioned that their mother called for help through their relatives and friends. When the police and social workers visited their home, it was already a week after the DV incident. They felt that the social workers merely wanted to repeatedly collect family information and were not helpful at all. As a result, these mothers kept complaining to their relatives and friends.

Overall, the findings of this study showed that neither the women nor the children who had witnessed DV were satisfied and had recovered from the process. In cultural terms, we found that the mother’s personal social network may play an important role in supporting post-DV life. The health-care, police, and social work systems may have room for improvement in the work they do for mothers experiencing DV and their children.

## 6. Limitations

The study was limited to 10 participants, which may lead to narrow perspectives from DV victims. Future studies could continue to focus on family experiences of DV and allocate more time to collecting data. Further research is needed to understand the influence of cultural characteristics (e.g., the cultural milieu of education in individualism vs. collectivism) on strategies for coping with DV. Furthermore, this study did not focus on how the victims adapted to their feelings after experiencing DV. The logical next step appears to be to examine this adaptation from the victims’ perspective.

## 7. Conclusions

To the best of our knowledge, this is the first study focusing on both mothers’ and their children’s DV experience. Our findings show that the women had to cope with not only the DV itself but also the negative consequences of the violence. The children experienced specific DV episodes and the consequences for family dynamics. Several recommendations are made based on the findings, including for the police, first-line health-care providers, and social workers. Suggestions are also made for the prevention of DV as well as for integrating the care provided to mothers experiencing DV and to their children.

## 8. Relevance to Clinical Practice

The findings of this study have several implications. (a) A well-publicized emergency telephone number may help reduce women’s fears, while restraining orders and protection of victims’ identities could avert potentially dangerous situations. (b) The police are recommended to implement high-level protective action, such as providing high-risk management, instead of only restraining orders. In addition, the police should protect the victim’s identity and provide the victim with an emergency telephone. (c) The first-line health-care providers should have professional responsibility and the ability to screen and detect cases and provide appropriate assistance and referral for treatment. (d) Children may require a professional team and a complete plan, which considers education, policing, social welfare, psychological counseling and psychiatric medicine, health assessment, and other professional areas. (e) In Taiwan, the DV Shelter Centers provide services and short-term placements for women who are victims of DV. The services further connect many resources for the women to support their living conditions until they can live independently and their children are provided better care. The functions of DV Shelter Centers should be more actively implemented. (f) The third and fifth measures are achievable as the primary preventative strategies for DV as the primary prevention can be achieved. These can include speeches in schools, parenting seminars in the community, and the government’s continuous communication of the harm and impact of DV on children through television and other media. (g) In Taiwan, some women’s foundations have launched parenting activities, such as art therapy, to help women and children suffering DV. Social workers can refer DV victims to these activities during their initial contact.

## Figures and Tables

**Table 1 healthcare-10-01556-t001:** Summary of themes and subthemes emerging from the interviews.

Participants	Theme	Subtheme
Women	Living with suffering	1.1 Physical pain
		1.2 Emotional distress and suffering from ambivalence
	2. Insecurity in daily life	2.1 Dealing with low self-esteem and self-comfort
		2.2 Heavy sense of loss
	3. Conformity in coping behaviors	3.1 Seeking social resources
		3.2 Living in the moment
Children	Barriers to learning and interaction with peers	1.1 Alienation from peers
		1.2 Learning gap led to frequent punishment
	2. A sense of threat to life	2.1 Monotonous home life
		2.2 Pessimistic thinking and low self-esteem

## Data Availability

Data sharing is not applicable to this article because this was a qualitative study.

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
