# Peer review of "Lived Experiences of Domestic Violence in Women and Their Children: A Phenomenological Study"

_healthcare, 2022, doi:10.3390/healthcare10081556_

Round 1

Reviewer 1 Report

The study explores women and their children’s lived experience of domestic violence (DV). I think the topic is coherent to the scope and aim of the journal and it could be of a great interest also for an international reader, I just suggest to read and add in the background the following paper: Nittari, G.; Sagaro, G.G.; Feola, A.; Scipioni, M.; Ricci, G.; Sirignano, A. First Surveillance of Violence against Women during COVID-19 Lockdown: Experience from “Niguarda” Hospital in Milan, Italy. Int. J. Environ. Res. Public Health 202118, 3801. https://doi.org/10.3390/ijerph18073801.

However, the methodology is clearly explained, and the manuscript is written according the guidelines of the journal.

Author Response

Dear Editor& Reviewer,

The authors sincerely appreciate for the opportunity to revise our paper. We have revised this paper by the reviewer’s suggestions and comments. Our responses have been listed in the table. Thank you.

With Best Regards,

Bih-O Lee, PhD, RN. Professor,

Dean of College of Nursing, Kaohsiung Medical University, Kaohsiung, Taiwan

Reviewer 2 Report

Dear writer(s) and Editor

Very interesting writing that could allow us to reaffirm the results of other investigations. Nevertheless, it seems to me that there are important changes to be made for this objective. Basically it is a question of complementing.

We find the need to be more explicit in relation to only three scientific aspects: the Statement of the Problem, Questions of Research and methodological and method issues.

Regarding the statement of the problem, it is very important to specify precisely and unequivocally: what does the contribution consist of? Apparently, there are no specific questions, about the originality, the specification of the thesis. It is required to contribute about empirical research to clearly understand this writing.

In the introduction, at the beginning of the writing, it is recommended to work more on the problem statement. Likewise, in the results of the investigation, equally, be very precise in the contributions of the investigation.

The objective presented: “The aim of this study was to explore the entire DV experience of women and children and use in-depth interviews to discuss the relationships and interactions between women and children suffering from DV”, it is too general, ubiquitous, and ambiguous.

Also, the matter of the Representativeness of the Sample is something that it is suggested to work on.

Just a name cannot only be given with respect to the community or social institution with which one works, it cannot only be said with respect to the identity of the study subjects: “Purposive sampling was used to select participants from a private community group called Women's New Knowledge Foundation (WNKF) in Taiwan.” In all sense, it’s a need to put the information in context. Where does WNKF come from, what is its institutional profile, who does it represent, what are its goals, its vision, sponsors? It seems to me that the precise methodological steps for determining the sample have not been followed. Ten people cannot be representative of Taiwan, of globalization, of whom? Why those subjects and not others, that age, that sexuality?

There is a lack of a clear connection between what the interviewees said and the measurement indicators. It is necessary to delve into the method of reducing the qualitative to the clusters, there is a lack of integrative coherence between the clusters or the categories and what was said by the informants.

The reference to a Phenomenology is worrying, because it appears without an academic School of inscription, without names of authors, without categories, logics and heuristics typical of that phenomenology. "Descriptive phenomenology" it is not a School, neither a methodology. It is recommended to be more specific in the content of the theoretical framework.

It also appears as an important limitation the fact of not studying, or not reporting as a variable in this writing, the cultural context, the social context, the space of reality. The emotional condition of the victim is another variable that should be considered as an inclusive criterion in the variables.

The results of all qualitative research only make sense through the relativization of both the singular and the general, that is, the true value of personal information is necessarily only in relation to the generality or the majority. Otherwise, it is not understood in terms of a real social population, of a real culture with respect to DV.

Observations on the method and methodology

It is recommended to specify what is the purpose of the research? Does it say that it is to impact public policies, only in Taiwan, or what is the specific political action to emulate? Is it to equate the globalist system of gender ideology with that of Taiwan? to see how much equates the problem between the global and Taiwan?

It would be very valuable to have some connecting statement between the results of the interviews and the general results of the research. Despite claiming the qualitative method of Phenomenology, its main advantage is not taken advantage of, which is the importance of the subjective.

For example, through the informants' sayings, abstract concepts produced from theory can be questioned. In any case, here it is necessary to present why are these subjects and not others? Those ages, that economic and political profile?

A personal language is a way of speaking, of explaining the ethos, it requires a connection between the agent, or informant, and the community to which he belongs. Thus, the samples have to be representative of a human conglomerate and need to be specified.

Likewise, it is necessary to specify what is meant by “a reflective journal was used to increase the richness and credibility of the data”. Even more so being a bibliographic citation to be clarified, but also when it says: “we invited two nursing professionals with wide experience of qualitative research to review and modify the classification of the findings.”

Another methodological flaw in which work is invited is that medical symptoms, mental health, must be reduced to those validated by international scales of prestige and world recognition, for example the ICD or the DSM.

Even more so due to the lack of information that is proposed to be completed here, a note that refers to non-determinism is very much needed here, because as other authors mention, there is no direct and simple relationship between what parents do and the emotional form of children. living off the children, especially their own intimate relationships.

Finally, some words about the research questions. It seems to me that the pertinent questions to contribute new knowledge are those that can be related to the concept of "intergenerational transmission of violence", a concept already widely applied, but very useful thinking in terms of what is taught at home. How is violence taught and transmitted from parents to children? How do children themselves live with and complement each other, or survive and overcome domestic violence? Why do some relatives, despite living with the same parents, may marry and others do not? Or does DV have to do with divorce?

Author Response

(The authors gave the same response as above.)

Round 2

Reviewer 2 Report

Dear Authors,

I expected more changes, but I think it has good points and statments to share. I just invite you to put you in the place of the reader, asking if your research can be replicate. We ought to be totally clear to our readers.

Thank you.